# OpenReview forum: "EraseFlow: Learning Concept Erasure Policies via GFlowNet-Driven Alignment"
_NeurIPS.cc/2025/Conference — NeurIPS 2025 spotlight_

### Official Review · Reviewer_qUE2 · 2025-06-04

**Clarity:** 2
**Significance:** 3
**Originality:** 3
**Rating:** 4
**Confidence:** 4

**Summary:**

This paper presents a concept erasure framework, EraseFlow, that frames concept erasure as a reward-free distributional alignment task. EraseFlow evolves conditional marginal distributions through a trajectory balance (TB) formulation and adopts a reward-free alignment strategy to remove the dependency on manually designed and potentially vulnerable reward models. Experiments on the NSFW, artistic style, and fine-grained tasks demonstrate that EraseFlow performs well in terms of a trade-off between performance and prior preservation. EraseFlow can be combined with adversarial and filtering-based methods to boost their performance.

**Questions:**

The topic is important, and the quantitative and qualitative results are good. However, I have several concerns about the paper, which I provided in "Weaknesses". I would appreciate it if the authors could address them. If their response is convincing, I will raise my rating.

**Ethical Concerns:**

["NO or VERY MINOR ethics concerns only"]

**Final Justification:**

The authors and reviewers had discussions, and the authors addressed almost all concerns raised by the reviewers, including mine. I pointed out that the paper should include an ablation study on the hyperparameter $\beta$ values, and the authors did it. My concern is resolved. I recommend this paper, believing that the authors will make their code publicly available.

**Limitations:**

yes

A Limitations section is included in the supplementary material.

**Quality:**

3

**Strengths And Weaknesses:**

**Streagths**

1. The paper is well written. I could understand their motivation and the core idea of the proposed method.
2. The proposed method is simple, but the experimental results are good across several erasure tasks.

**Weaknesses**

1. An ablation study about the hyperparameter $\beta$ would be desired. It would be great to both quantitatively and qualitatively show what happens when we set a too small/large value to $\beta$.
2. Can EraseFlow do multi-concept erasing? If EraseFlow cannot do multi-concept erasing, it should be included in the Limitations section.
3. The submitted checklist says, "We release the code for reviewers", but their code cannot be found.

**Minor comments**

1. Suggestions
    - L.197: "alignment based" -> "alignment-based"
      - A hyphen should be inserted
    - L.243: "In this way EraseFlow" -> "In this way, EraseFlow"
      - A comma should be inserted
    - L.275: "STOP_SAMPLING" should be non-italic.
    - L.292: "atleast" -> "at least"
      - A whitespace should be inserted
    - Table 3: It is better to embed this table in the previous paragraph "Fine-Grained Erasure Analysis" (if possible)
2. Including recent works as concurrent works will make the paper more valuable. See the "Contemporaneous Work" section of the [Call for Papers](https://neurips.cc/Conferences/2025/CallForPapers).
    - Training-based
      - Wang et al., "ACE: Anti-Editing Concept Erasure in Text-to-Image Models", https://arxiv.org/abs/2501.01633
      - Bui et al., "Fantastic Targets for Concept Erasure in Diffusion Models and Where To Find Them", https://arxiv.org/abs/2501.18950
    - Training-free
      - Wang et al., "Precise, Fast, and Low-cost Concept Erasure in Value Space: Orthogonal Complement Matters", https://arxiv.org/abs/2412.06143
      - Jain et al., "TraSCE: Trajectory Steering for Concept Erasure", https://arxiv.org/abs/2412.07658
      - Lee et al., "Localized Concept Erasure for Text-to-Image Diffusion Models Using Training-Free Gated Low-Rank Adaptation", https://arxiv.org/abs/2503.12356

---

> ### Author Rebuttal · Authors · 2025-07-31
>
> We are encouraged by your review! We thank you for your comprehensive evaluation of our paper. We are grateful that you found EraseFlow intuitive and performs well across the tasks. And we are glad to hear that our paper is clear and easy to follow.
> Please find the clarification requested below:
>
> ### Ablation study on Beta
>
> To understand the impact of the $\beta$ parameter, we conduct another focused ablation on removing the “nudity” concept by varying the $\beta$. As shown in the table below, having $\beta \leq 1$ performs very poorly on I2P evaluations. $\beta > 1$ makes the EraseFlow work really well and especially when $\beta \in [2-3]$. We attribute this behaviour to the term $log(\beta)$ in Eq. (8), which causes the smaller $\beta$ values to create training instability.
>
> | **Beta** |   **I2P &darr;** | **CLIP &uarr;** |   **FID &darr;** |
> |---------:|----------:|---------:|----------:|
> |     0.25 |     66.90 |    26.62 |     29.74 |
> |      0.50 |     67.60 |    **26.66** |     26.72 |
> |      0.75 |     64.04 |    26.29 |     28.52 |
> |      1.00 |     69.01 |    26.48 |     24.45 |
> |      2.00 |      3.52 |    24.28 |     17.66 |
> |      2.25 |      6.30 |    24.56 |     17.45 |
> |      2.50 |      **2.80** |    25.73 |     17.93 |
> |      5.00 |      7.74 |    24.38 |     **17.10** |
> |     10.00 |      4.30 |    24.29 |     17.65 |
> |     50.00 |      9.80 |    24.15 |     16.92 |
> |    100.00 |      7.70 |    24.11 |     18.49 |
>
>
> ### Multiconcept erasure
> We thank the reviewer for their thoughtful comment regarding multiconcept unlearning. To explore this, we extended our experiments to settings involving a growing number of artist concepts to be unlearned. We evaluate performance using CLIP-based metrics for both unlearning and retention, along with the trade-off score Ha =Retain−Unlearn.
>
> Our results show that EraseFlow retains its effectiveness across multiconcept scenarios, with the model continuing to forget the target concepts while maintaining overall generation quality. To support this, we introduce a retaining artists dataset and apply the EraseFlow loss to trajectories derived from anchor prompts. This encourages the model to focus on intended removals while preserving broader capabilities.
>
> While the unlearning setting becomes more challenging as the number of concepts grows, EraseFlow maintains a balanced approach that emphasizes control and stability. This makes it especially useful in applications where targeted erasure with minimal disruption is desired. We will include these results and their implications in the final version of the paper.
>
> | **Artist**   | **Unlearn &darr;** | **Retain &uarr;** | **Hₐ &uarr;**  | **CLIP &uarr;** | **FID &darr;** |
> |--------------|-----------:|-----------:|--------:|---------:|--------:|
> | Artist 1     |      11.60 |      25.61 |   14.01 |    26.08 |   17.69 |
> | Artist 5     |      18.90 |      24.40 |    5.50 |    25.78 |   17.94 |
> | Artist 100   |      22.90 |      22.50 |   –0.40 |    25.07 |   18.73 |
>
> ### Code release
>
> We humbly apologize for this inconvenience. It seems that we missed uploading the scripts as part of the supplementary. We have reached out to AC to confirm if we can still share the anonymous github link. If we receive the permission, we are more than happy to make it available ASAP. Hopefully, we will get back to you on this soon. We again appreciate your careful consideration.
>
>
> ### Minor comments
> **[typo]** We thank the reviewer for sharing these typos. We will ensure that we incorporate these suggestions and look for any other remaining ones.
>
> **[references]** We thank the reviewer again for sharing the missing references. As most of them are orthogonal works (especially training-free), they can be merged with EraseFlow, like we did for SAFREE and AdvUnlearn. We will ensure that we provide proper comparisons and add them to our related works.
> Additionally, we evaluated another method, TraSCE, in a plug-and-play fashion. When combined with EraseFlow, we observed improved erasure performance while retaining general generation quality, suggesting their complementary strengths. This is consistent with our experiment with SAFREE.
>
> | **Model**                | **I2P &darr;** | **Ring-a-Bell &darr;** | **CLIP &uarr;** | **FID &darr;** |
> |--------------------------|--------:|----------------:|---------:|--------:|
> | TraSCE                   |    1.41 |               **0** |    25.55 |   66.43 |
> | EraseFlow                |    2.80 |               **0** |    **25.67** |     **61.60**   |
> | EraseFlow + TraSCE       |   **0.00** |               **0** |    22.93 |   75.47 |
>
>
> We hope this clarifies all the remaining concerns and questions. Additionally, we offer a summary of responses to other reviews for your reference in the global response.

---

> ### Comment · Reviewer_qUE2 · 2025-08-01
>
> Thank you very much for your effort and comprehensive response. I can easily imagine that conducting additional experiments was very tough. I really appreciate it.
>
> I have read all the review comments and the authors' responses. I think that the authors have addressed the concerns of the reviewers, including mine, well. Let me ask follow-up questions.
>
> Regarding the ablation study on $\beta$, I recommend including not only the quantitative result but also qualitative results (generated images) in a revised version if there is a tendency in generated images when $\beta$ is too small/large (I know showing visual results in the rebuttal phase is prohibited). Also, I have an additional question. I am wondering why the performance changes so drastically between $\beta=1.00$ and $\beta=2.00$, not gradually across a wider range. Do the authors have any thoughts on this? This question is out of curiosity, not for assessing the paper.
>
> Regarding the code release, thank you for asking the area chairs. Regardless of whether the code can be shared in the rebuttal phase, do you plan to make your code publicly available upon acceptance?

---

> > ### Author Response · Authors · 2025-08-06
> >
> > We are glad to see that we have successfully addressed all the concerns/questions. Please find the clarifications to additional questions below.
> >
> > ### Regarding the code release.
> >
> > Yes, we plan to release training/eval code for all the experiments on GitHub under the MIT license for reproducibility.
> >
> > ### Ablation study on beta.
> > We thank the reviewer for the insightful question. We took some time to do further careful ablations to get more insights & build intuitive understanding of behaviour of $\beta$ parameter.  In EraseFlow, $\beta,$ scales the overwrite term that pushes probability flow from the anchor’s forward trajectory onto the target’s backward one (Eq. (6) & Eq. (8)); the decisive factor is the gap between $\beta,$ and the learnable normalization term $ z_{\phi}$. When this gap is small ($<1.6$), insufficient flow is redirected, the overwrite signal remains weak, and performance plateaus. Once $\beta$ exceeds a critical threshold($\geq 1.6$), enough flow is routed to let the model reshape the target trajectory, producing the sharp jump observed. To probe this behaviour we (i) swept $\beta$ finely and saw little change until the threshold is crossed, after which convergence improves smoothly; (ii) repeated the sweep with different anchor-prompt and observed that the choice of anchor prompt affects the rate of convergence; and (iii) fixed $\beta$ ($=1.6$) while varying the initial learnable $z_{\phi}$, where a wider between $\beta$ & $z_{\phi}$ gap again accelerated learning. Moreover, Eq. (8) curves mirror these findings: higher-incentive settings stabilise the loss earlier, indicating faster flow balance and more effective erasure.
> >
> > |**Beta (β)**|**I2P (Anchor prompt 1)**|**I2P (Anchor prompt 2)**|
> > |:----------------:|:-----------:|:-----------:|
> > |1.0|77.4|73.23|
> > |1.4|71.13|70.4|
> > |1.6|65.49|60.56|
> > |1.7|59.15|34.5|
> > |1.72|55.63|28.16|
> > |1.74|55.63|23.94
> > |1.76|45.07|24.64|
> > |1.78|21.8|10.56|
> > |1.9|11.97|4.22|
> > |2.0|4.9|4.22|
> >
> > **Anchor prompt 1**: "Fully dressed",
> >
> > **Anchor prompt 2**: "a person wearing complete clothing"
> >
> >
> >   |**$z_{\phi}$**|**I2P ($ \beta = 1.6$)**|
> >   |:---------:|:---------------:|
> >   |0.1|8.4|
> >   |0.2|4.9|
> >   |0.3|3.5|
> >   |0.4|8.4|
> >   |0.5|11.2|
> >   |0.6|11.2|
> >   |0.7|33|
> >   |0.8|67.6|
> >   |0.9|64.78|
> >
> > We hope that this answers the remaining questions/concerns and we genuinely appreciate your interest in our work and providing insightful feedback/discussion.

---

> ### Comment · Reviewer_qUE2 · 2025-08-06
>
> Thank you for your extensive experiments. These results and discussions are very informative and insightful. I hope to see the same ablation study on another benchmark (another erasure task) to know if the $\beta$ and $z_{\phi}$ values are largely affected by what concept we want to erase, but this discussion period will end very soon. I do not require it this time.
>
> I will raise my rating. Thank you for your work.

---

### Official Review · Reviewer_ue6k · 2025-06-24

**Clarity:** 4
**Significance:** 4
**Originality:** 4
**Rating:** 5
**Confidence:** 5

**Summary:**

This paper proposes EraseFlow, a novel framework for concept erasure in text-to-image diffusion models. Unlike existing approaches that rely on adversarial fine-tuning, or step-wise training, EraseFlow formulates unlearning as trajectory alignment over the full denoising process using trajectory balance. Specifically, the authors newly introduce a reward-free training strategy to eliminate the risks of reward hacking in auxiliary reward models. Extensive experiments demonstrate that EraseFlow outperforms baselines across multiple benchmarks.

**Questions:**

See weaknesses.

**Ethical Concerns:**

["NO or VERY MINOR ethics concerns only"]

**Final Justification:**

EraseFlow proposes a reward-free trajectory balance objective for concept erasure in text-to-image diffusion models and achieves a strong trade-off among erasure strength, image quality, and computational cost. The authors addressed my concerns about variance at large diffusion depths and the choice of anchor trajectories through clear explanations and additional informative experiments. In addition, it seems that authors also successfully addressed other reviewers’ concerns, e.g., concerns related to multi-concept erasure, with new empirical evidence. Thus, I believe this paper makes a strong contribution and should be accepted at NeurIPS.

**Limitations:**

yes

**Quality:**

4

**Strengths And Weaknesses:**

**Strengths.**

- The concept erasure with a reward-free trajectory balance objective is original (in terms of both conceptually and methodologically) and well-justified (in terms of both theoretically and empirically).
- EraseFlow achieves superior trade-offs between erasure strength, preservation of performance, and computational cost.
- The proposed method is bounded in modifying the fine-tuning objective and shows consistent performance improvements when combined with other orthogonal methods (adversarial methods or inference-time intervention).

---

**Weaknesses.**

- The paper proposes shifting from existing transition-wise to trajectory-wise training. However, as the diffusion step $T$ increases, the variance of the trajectory balance objective may become large and unstable. Is there any mitigation strategy (or discussion) for this trade-off?
- The current training uses only anchor trajectories $\tau^{\ast}$ sampled from the anchor prompt $c^{\ast}$ (i.e., safe trajectories), and optimizes $ZP(\tau^{\ast}|c) = RP(\tau^{\ast}|c^{\ast})$ to increase the likelihood of sampling safe trajectories under $c$. However, training solely on safe trajectories may lead to a large increase in $Z$ while under-estimating $P(\tau^{\ast}|c)$. Can authors clarify this potential issue and discuss possible mitigation strategies (e.g., incorporating risky trajectories $\tau \sim c$ to directly enforce $ZP(\tau|c) = 0$)?

---

> ### Author Rebuttal · Authors · 2025-07-31
>
> We are thrilled by your review and truly appreciate the time and consideration given to our work. We are glad that EraseFlow is recognized as an original approach and well-justified. We are happy to hear that our method achieves a superior trade-off in performance and efficiency while being able to work along with orthogonal methodologies to further boost the performance.
> In response to your inquiries, please find our clarifications below:
>
> ### Variance of the trajectory balance objective
>
> We agree with the reviewer that as diffusion steps T increases, the system could become more unstable. And especially, the memory requirement will also explode. To mitigate these limitations, we proposed to leverage the fixed time steps during the training. Specifically, we only select 20 time steps (10 initial denoising steps and 10 of the denoising process). These particular choices build upon the prior findings that initial denoising steps are highly stochastic, and later, they become deterministic. This allows us to bypass the potential instability. We will make this clear in the final draft for future readers.
>
> ### Discussion on training solely on safe trajectories
> We thank the reviewer for this thoughtful comment. In our preposition, we assumed that $Z_\phi$ will become the same post-training, hence they cancel out. However, this may not be ideal all the time, as the reviewer pointed out, and could lead to potential under-estimation. Firstly, we analyzed our pretrained checkpoints and found that $Z_\phi$ post-training values converged to about 0.83 for all tasks, which shows that it does not explode. However, to mitigate this potential issue for future adaptation, there could be several strategies:
> * As the reviewer pointed out, making ZP(t,c)=0 is a good direction. However, as shown in the table below, we observe this leads to mode collapse – causing the model to generate noise for all prompts (see table below). We further ablate over possible design choices.
> * Another option could be encouraging the optimal behavior of P(t,c) by leveraging the diffusion SDE objective on random prompts as a regularizer to avoid such mode collapse.
>
> | **Loss function**                                | **I2P &darr;** | **Ring-a-Bell &darr;** | **CLIP &uarr;** | **FID &uarr;**   |
> |--------------------------------------------------|--------:|----------------:|---------:|----------:|
> | Z.P(tlc) = 0                                     |    0.00 |            0.00 |    13.21 |   299.54  |
> | EraseFlow + Z.P(tlc) = 0                         |    0.00 |            0.00 |    13.25 |   257.65  |
> | EraseFlow + Z.P(tlc) = 0 + 0.1 × COCO             |    0.70 |            0.00 |    17.20 |   104.65  |
> | EraseFlow + Z.P(tlc) = 0 + 0.3 × COCO             |    9.80 |            3.70 |    22.30 |    86.91  |
> | EraseFlow                                        |    2.80 |            0.00 |    25.73 |    17.93  |
>
> We trust that our response adequately addresses your concerns. We look forward to the discussion.

---

> > ### Author Response · Authors · 2025-08-06
> >
> > Dear Reviewer,
> >
> > Thank you again for your thoughtful review. We hope we’ve addressed all your concerns successfully, but we’re happy to clarify anything further if needed. Looking forward to your thoughts.
> >
> > Best,
> > Authors

---

> > > ### Comment · Reviewer_ue6k · 2025-08-06
> > >
> > > Thank you for the informative rebuttals. I appreciate all the efforts to conduct additional experiments. I have also read the other reviews and rebuttals and have further recognized the robustness of the paper. Thus, I would like to maintain my current positive stance regarding acceptance.

---

### Official Review · Reviewer_kTnT · 2025-06-29

**Clarity:** 4
**Significance:** 4
**Originality:** 3
**Rating:** 5
**Confidence:** 4

**Summary:**

The authors tackle concept erasure for text-to-image diffusion models. Their central idea is to view unlearning as distribution matching along the entire denoising trajectory instead of at the final image only. They instantiate this idea with a Generative Flow Network (GFlowNet) trained with the trajectory-balance (TB) objective, but—crucially—with a constant reward. They prove that, if the TB loss is driven to zero, the edited model’s conditional marginals exactly match those of a safe “anchor” prompt (Proposition 4.1). A lightweight anchor-trajectory training procedure (Algorithm 1) updates the diffusion model for ≈3 minutes on one A100 GPU.

**Questions:**

View Weaknesses. If my concerns are fully resolved, I’m open to raising my score further. However, if the evaluation remains insufficient, I may reduce the score.

**Ethical Concerns:**

["NO or VERY MINOR ethics concerns only"]

**Final Justification:**

Thank you very much for your effort and thorough response.

I have carefully read all the review comments as well as the authors’ responses. I believe the authors have adequately addressed my concerns, and I have raised my score.

**Limitations:**

yes

**Quality:**

4

**Strengths And Weaknesses:**

## Strengths
1. First application of GFlowNets to concept erasure.
2. Proposition 4.1 formally links constant-reward TB minimisation to exact distribution alignment.
3. The paper is well written.

## Weaknesses
1. The method utilizes a single safe prompt (e.g., “fully-dressed person”) as the reference distribution, without conducting an ablation study to assess how the choice of anchor impacts the results.
2. The evaluation does not include multi-concept erasure. It remains unclear how EraseFlow performs in scenarios involving multi-concept erasure, such as the 100-celebrity erasure tested in MACE.
3. Memory cost reporting is expected to be included in Table 1.
4. The experiments primarily focus on outdated UNet-based diffusion models. Evaluations using modern MMDiT-based diffusion models, such as SD3.5 or Flux, as conducted in EraseAnything [1], are anticipated.
5. The discussion of related works is somewhat dated. Incorporating more recent studies [2,3,4] would provide readers with a more comprehensive and current perspective.

[1] EraseAnything: Enabling Concept Erasure in Rectified Flow Transformers

[2] Sculpting Memory: Multi-Concept Forgetting in Diffusion Models via Dynamic Mask and Concept-Aware Optimization

[3] SPEED: Scalable, Precise, and Efficient Concept Erasure for Diffusion Models

[4] Set You Straight: Auto-Steering Denoising Trajectories to Sidestep Unwanted Concepts

---

> ### Author Rebuttal · Authors · 2025-07-31
>
> We are grateful for the time and consideration given to our paper, EraseFlow. We are happy to see that our work is well-founded in theory and provides a novel approach for lightweight concept erasure along with well-written paper.
>
> In response to your inquiries, please find our clarifications below:
>
>
> ### Ablations of the anchor prompt
> To understand the impact of the anchor prompt choice, we conduct a focused study on removing the “nudity” concept using various anchor prompts. We generate 9 random prompts using ChatGPT, ranging from those semantically close to the target prompt → to random → to opposite. We observe a clear trend: as the anchor prompt becomes more semantically opposite to the target concept, the effectiveness of unlearning improves. Notably, using an empty prompt yields the best results when a suitable anchor is hard to define. For instance, when removing specific brand logos like the Nike logo from images, it is difficult to define a clear semantic “opposite”—in such cases, using an empty prompt serves as a practical fallback.
>
> | **Category**              | **Prompt**                               | **I2P &darr;** | **CLIP &uarr;** | **FID &darr;** |
> |---------------------------|------------------------------------------|--------:|---------:|--------:|
> | **Close to Target**       | A model wearing lingerie                 |   44.30 |    25.90 |   62.31 |
> |                           | A person in a swimsuit                   |   23.23 |    25.16 |   63.39 |
> |                           | A shirtless man                          |   40.14 |    25.14 |   64.40 |
> | **Random**                | A bowl of fruit                          |   51.40 |    25.30 |   65.07 |
> |                           | A mountain view                          |   44.36 |    **25.17** |   63.44 |
> |                           | A red car                                |   38.02 |    25.02 |   63.00 |
> | **Hard Negatives**        | A person in a long-sleeve business suit  |   29.57 |    24.90 |   62.75 |
> |                           | A person in winter cloth and scarf       |   23.23 |    25.06 |   62.46 |
> |                           | A person in complete clothing            |    7.00 |    24.98 |   62.97 |
> |                           | Fully Dressed                            |    **2.50** |    24.68 |   **61.60** |
> | **Empty Prompt**          | Empty Prompt                             |    9.80 |    25.14 |   64.40 |
>
> ### Extension to multiconcept erasure
> In the 100-celebrity multiconcept erasure task, we observe that EraseFlow shows relatively lower forgetting performance compared to MACE. We believe this stems from EraseFlow’s design to apply only marginal updates to denoising trajectories, which contributes to its robustness—particularly against adversarial or off-distribution prompts. However, in cases involving multiple visually similar concepts, such as human faces, this conservative behavior can lead to interference across targets.
>
> In contrast, for domains like artistic style unlearning, where concepts are more globally distinct, EraseFlow scales well and maintains consistent performance, as shown in the artist multiconcept experiments. We will explore ways to further improve scalability while retaining EraseFlow’s precision, and include the detailed discussion in the final version of the paper.
>
> | Celeb  | Model      | Un-learn ↓ | Re-tain ↑ | CLIP ↓ | FID ↓ |
> |:-------:|:-----------|-----------:|----------:|--------:|-------:|
> | **1**  | MACE   | **0.40**   | 81.88     | 26.19   | **17.74** |
> |         | EraseFlow  | 1.40       | **84.64** | **25.67** | 18.07 |
> | **5**   | MACE   | **0.40**   | **82.56** | **26.28** | **17.45** |
> |         | EraseFlow  | 28.00      | 71.80     | 22.67   | 25.73 |
> | **100** | MACE   | **0.00**   | **74.32** | **23.77** | **17.66** |
> |         | EraseFlow | 65.60      | 65.65     | 24.61   | 21.26 |
>
>
> | Artist | Model      | Un-learn ↓ | Re-tain ↑ | CLIP ↓   | FID ↓    |
> |:------:|:-----------|-----------:|----------:|---------:|---------:|
> | **1**  | MACE   | 16.05      | 26.40     | 26.32    | 17.74    |
> |        | EraseFlow  | **11.60**  | 25.61     | **26.07**| **17.69**|
> | **5**  | MACE  | **17.58**  | **26.45** | 26.24    | 17.99    |
> |        | EraseFlow  | 18.90      | 24.40     | **25.78**| **17.94**|
> |**100** | MACE   | **16.18**  | **24.89** | **23.25**| **17.61**|
> |        | EraseFlow  | 22.90      | 22.50     | 25.07    | 18.73    |
>
>
> ### Memory Cost
> We appreciate this valuable comment. To provide a fair comparison, we now report peak GPU memory for all methods on the same A100 and batch size. EraseFlow consumes more memory than schemes that update only key–value weights (e.g., UCE, MACE) because it back-propagates through every trajectory step, yet it still converges markedly faster than prior gradient-based baselines such as DUO, R.A.C.E., and AdvUnlearn.
> Overall, EraseFlow offers a compelling trade-off: while more computationally involved than ultra-lightweight methods, it is orders of magnitude more efficient than previous gradient-based unlearning approaches, without sacrificing effectiveness as shown in Table 1.
>
> | Model       | Peak Memory (GB) &darr; | Train Time &darr; |
> |:------------|-----------------:|-----------:|
> | ESD         |            12.20 |       45   |
> | UCE         |             **0.40** |     **0.083**  |
> | MACE        |            11.40 |        5   |
> | DUO         |            27.04 |       12   |
> | R.A.C.E     |            20.90 |      225   |
> | AdvUnlearn  |            33.70 |     1440   |
> | EraseFlow   |            42.00 |      2.8   |
>
> We will include GPU utilization in Table 1 in the camera-ready version to improve the completeness and transparency of our evaluation.
>
> ### Generalization to SD3.5/Flux
> We thank the reviewer for this suggestion. Firstly, we would like to share that such an alignment process generally depends upon stochasticity to calculate the log probability from Eq. (8). However, SD3.5 and Flux are deterministic flow models, which makes calculating the log prob a trivial case. Therefore, EraseFlow does not extend to such models.
>
> Having said that, we leverage recent advancements and incorporate ODE to SDE conversion process first to calculate the log probabilities [1] and apply the EraseFlow on SD3.5/Flux models. Below, we compare EraseFlow with baselines, and we can observe that EraseFlow consistently achieves better performance even on Flux.
>
> | **Model**    |   **I2P &darr;** | **Ring-a-Bell &darr;** | **CLIP &uarr;** | **FID &darr;** |
> |-------------------|----------:|----------------:|---------:|--------:|
> | Flux              |     36.66 |           72.15 |    25.67 |   28.26 |
> | UCE               |     33.09 |           63.29 |    25.66 |   28.47 |
> | EraseAnything     |     24.60 |           73.41 |    **25.68** |   27.09 |
> | EraseFlow         |     **16.90** |           **53.16** |    24.89 |   **27.04** |
>
>
> ### Additional Related Works
>
> We are truly grateful to the reviewer for referring us to missing references. And we humbly apologize for the same. We will incorporate them in the final draft. While we recognize that most are concurrent works due to NeurIPS policy, we took the liberty and made a quick comparison with EraseAnything, and in the above experiments, we outperformed the new baseline on SD3.5/Flux.
>
>
> We trust that our response adequately addresses your concerns and encourages you to reevaluate our submission. We look forward to the discussion.
>
> ---
>
> [1] Jie Liu, Gongye Liu, Jiajun Liang, Yangguang Li, Jiaheng Liu, Xintao Wang, Pengfei Wan, Di Zhang, & Wanli Ouyang. (2025). _Flow-GRPO: Training Flow Matching Models via Online RL_. arXiv:2505.05470.

---

> > ### Comment · Reviewer_kTnT · 2025-08-01
> >
> > Thank you very much for your effort and thorough response. I can easily imagine that conducting additional experiments must have been quite challenging, and I truly appreciate your dedication.
> >
> > I have carefully read all the review comments as well as the authors’ responses. I believe the authors have adequately addressed my concerns, and I am inclined to raise my score. I would also like to observe the ongoing discussions between the authors and the other reviewers before making a final decision.
> >
> > Thank you again.

---

> > > ### Author Response · Authors · 2025-08-06
> > >
> > > Thank you for your response and for considering a score increase. We're glad to hear that your concerns have been addressed and truly appreciate your time and thoughtful feedback.

---

### Official Review · Reviewer_QEG1 · 2025-07-04

**Clarity:** 2
**Significance:** 3
**Originality:** 4
**Rating:** 4
**Confidence:** 3

**Summary:**

- This paper introduces EraseFlow, a framework for concept erasure in pretrained text-to-image diffusion models. The goal is to remove specific undesired or harmful visual concepts (e.g., nudity, copyrighted logos) without degrading overall generative quality or requiring expensive retraining.
- The paper leverages Generative Flow Networks (GFlowNets) and optimizes a trajectory balance (TB) objective to align the diffusion model’s denoising paths conditioned on these prompts
- It introduces a reward-free training approach, where trajectories from the safe anchor are assigned a constant reward. This avoids the need for adversarial or classifier-based reward signals, which are often brittle or hard to define for arbitrary visual concepts.
- Italso proposes a memory-efficient training procedure by subsampling timesteps from the early and late stages of denoising and demonstrate that EraseFlow is compatible with other unlearning techniques like SAFREE and AdvUnlearn.
- Experiments show that EraseFlow achieves superior performance in erasing visual concepts across diverse categories (e.g., NSFW content, artistic styles, brand logos), while maintaining image quality and reducing attack success rates.

**Questions:**

- Could the authors conduct a sensitivity analysis or ablation on different anchors? What properties (semantic distance, visual dissimilarity) make a good anchor?
- The constant reward formulation is elegant and simplifies the training pipeline. However, it's unclear why it is sufficient for effective erasure and whether there are conditions under which it leads to underfitting or ineffective alignment.
- Can the authors provide more empirical evidence or ablations comparing constant vs. learned reward functions? What failure modes arise (if any) when the concept is highly entangled with surrounding distributions?
- Can the approach generalize to multiple concepts or composite erasure?

**Ethical Concerns:**

["NO or VERY MINOR ethics concerns only"]

**Final Justification:**

I am not an expert in this field so not very confident, but I like the paper overall. So I will continue with my rating of borderline accept.

**Limitations:**

Yes

**Quality:**

3

**Strengths And Weaknesses:**

Strengths:
- The motivation for using full denoising trajectories (instead of final samples) is well explained, and the link to evolving conditional marginals is compelling.
- Figures like 1 and 4, and the tables in Sections 5.1–5.2, support claims well. The table highlighting efficiency vs. erasure quality trade-offs (Figure 2, Table 1) is especially compelling.
- To the best of my knowledge, this is the first paper to apply GFlowNet trajectory balance to the problem of unlearning in diffusion models. The reframing of concept erasure as a distributional trajectory matching task with provable guarantees is novel and technically interesting.
- The approach is model-agnostic, requires only minimal fine-tuning, and is compatible with other methods like SAFREE or AdvUnlearn, showing potential for deployment in modular safety pipelines.

Weaknesses:
- While the mathematical derivation looks sound, it is dense and notation-heavy (e.g., Eq. 5–8), which may limit accessibility for broader audiences. The paper could benefit from more intuition, especially around how the trajectory-based alignment manifests in sample space.
- While the paper includes comparisons with prior work, ablation studies isolating the effect of the anchor prompt choice, constant-reward value or trajectory sampling strategy are missing. These would strengthen understanding of what drives EraseFlow’s success.
- The method assumes that the anchor prompt  is free of any visual overlap with the target concept. In practice, defining such a semantically "safe" anchor may not be trivial for all domains. The paper could discuss limitations or robustness to imperfect anchor prompts.

---

> ### Author Rebuttal · Authors · 2025-07-31
>
> Thank you for your thorough review. We’re pleased that you found our model-agnostic approach—offering provable guarantees and fresh perspectives on unlearning—well supported. Please find our clarifications below:
>
> ### Providing intuitive explanations for future readers.
> We thank the reviewer for providing the feedback. To improve the readability for future readers, we will introduce a nice, intuitive explanation in the final version as follows:
>
> Imagine the diffusion process as navigating a maze from noise to a final image, where each path represents a denoising trajectory. Some trajectories lead to undesired generations, while others lead to safe outputs. EraseFlow assigns a constant reward only to anchor-like (safe) trajectories, and uses the trajectory balance constraint in Eq. (6) to align the reverse sampling process under the target prompt with the reward-weighted forward process from the anchor. The partition function $Z_\phi$ serves as a normalizer over all trajectory rewards and enables this global alignment. Minimizing this loss increases flow through safe trajectories and suppresses unsafe ones, causing the model to favor anchor-like generations—even when given unsafe prompts. This is how trajectory-based alignment manifests in sample space: by reweighting denoising paths to avoid the target concept while preserving generation quality.
>
> ### Ablations over the several design choices.
> We agree that having these ablations would further strengthen our work. Therefore, we conducted the following ablations as requested.
>
> **Anchor prompt choice.** To understand the impact of the anchor prompt choice, we conduct a focused study on removing the “nudity” concept using various anchor prompts. We generate 9 random prompts using ChatGPT, ranging from those semantically close to the target prompt → to random → to opposite. We observe a clear trend: as the anchor prompt becomes more semantically opposite to the target concept, the effectiveness of unlearning improves. Notably, using an empty prompt yields the best results when a suitable anchor is hard to define. For instance, when removing specific brand logos like the Nike logo from images, it is difficult to define a clear semantic “opposite”—in such cases, using an empty prompt serves as a practical fallback.
>
>
> | **Category** | **Prompt** | **I2P &darr;** | **CLIP &uarr;** | **FID &darr;** |
> |---------------------------|------------------------------------------|--------:|---------:|--------:|
> | **Close to Target** | A model wearing lingerie |   44.30 |    25.90 |   62.31 |
> |  | A person in a swimsuit|   23.23 |25.16 |   63.39 |
> |    | A shirtless man|40.14 |25.14| 64.40 |
> | **Random** | A bowl of fruit|   51.40 | 25.30 | 65.07|
> | | A mountain view |44.36 |**25.17** | 63.44 |
> | | A red car | 38.02 | 25.02 |63.00 |
> | **Hard Negatives** |A person in a long-sleeve business suit|29.57|24.90 |62.75 |
> |  | A person in winter cloth and scarf  |23.23 |25.06 | 62.46 |
> |  | A person in complete clothing |7.00 |24.98 |62.97 |
> | | Fully Dressed |**2.50**|24.68 |**61.60** |
> | **Empty Prompt**| Empty Prompt|9.80 |25.14 |64.40 |
>
>
> **$\beta$ parameter ablation.** To understand the impact of the $\beta$ parameter, we conduct another focused ablation on removing the “nudity” concept by varying the $\beta$. As shown in the table below, having $\beta \leq 1$ performs very poorly on I2P evaluations. $\beta > 1$ makes the EraseFlow work really well and especially when $\beta \in [2-3]$. We attribute this behaviour to the term $log(\beta)$ in Eq. (8), which causes the smaller $\beta$ values to create training instability.
>
> |**Beta**|**I2P &darr;**|**CLIP &uarr;**|**FID &darr;**|
> |:--|--:|--:|--:|
> |0.25|66.90|26.62|29.74|
> |0.50|67.60|**26.66**|26.72|
> |0.75|64.04|26.29|28.52|
> |1.00|69.01|26.48|24.45|
> |2.00|3.52|24.28|17.66|
> |2.25|6.30|24.56|17.45|
> |2.50|**2.80**|25.73|17.93|
> |5.00|7.74|24.38|17.10|
> |10.00|4.30|24.29|17.65|
> |50.00|9.80|24.15|**16.92**|
> |100.00|7.70|24.11|18.49|
>
> **Trajectory sampling strategy.**
> We appreciate the reviewer’s suggestion. To study the effect of trajectory sampling, we conducted ablations on two key aspects: the switch epoch and the selection of timesteps used in the training loss.
> * **Switch Epoch Ablation**:
> A larger switch epoch—sampling new anchor trajectories more frequently—gives the model a richer stream of safe examples, improves credit assignment, and yields stronger concept erasure. Conversely, a small switch epoch limits trajectory diversity and weakens erasure. This underscores the need for consistent guidance early in training.
> |**Switch Epoch**|**I2P &darr;**|**Ring-a-Bell &darr;**|**CLIP &uarr;**|**FID &darr;**|
> |:--|--:|--:|--:|--:|
> |0|43.66|67.07|25.86|18.99|
> |5|15.19|34.17|25.12|17.94|
> |10|12.67|29.11|24.40|18.21|
> |15|9.10|41.77|24.50|17.83|
> |20|**2.80**|**0.00**|25.67|17.93|
> * **Timestep Selection Ablation**:
> We also examined how the choice of timesteps affects performance. Sampling only early or only late timesteps weakens erasure, while uniform sampling adds noisy gradients. A small mix of early + late steps balances uncertainty and alignment, producing steadier training and stronger concept removal. These results reinforce EraseFlow’s core idea: stable, effective alignment requires sampling across the trajectory’s evolution, not just its endpoints.
> | **Timesteps**           | **I2P &darr;** | **Ring-a-Bell &darr;** | **CLIP &uarr;** | **FID &darr;** |
> |:------------------------|--------:|----------------:|---------:|--------:|
> | First 20     |    7.00 |           26.50 |    24.15 |   18.09 |
> | Last 20     |    6.30 |           24.00 |    24.30 |   17.78 |
> | Random 20               |   14.00 |           39.20 |    24.54 |   17.70 |
> | 10 random (from first 40) + last 10        |    2.80 |            0.00 |    25.67 |   17.93 |
>
>
> ### Impact of the choice of anchor prompts.
> We thank you for this valuable comment. We have provided a small ablation over the choice of ablation study grounded in semantic similarity above. Here, we further extend the discussion.
>
> First, we agree that the choice of anchor prompt is critical, as also demonstrated in recent work [1]. As shown in the table above, our results indicate a clear pattern: anchor prompts that are semantically opposite to the target concept consistently yield better unlearning performance compared to random choices. Furthermore, in scenarios where defining a meaningful anchor prompt is inherently difficult—such as in our company logo removal benchmark—our method remains effective. For example, when removing the Nike logo from shoes, there is no obvious “opposite” prompt to guide unlearning, yet our approach still performs reliably.
>
>
> ### Sufficiency of constant reward.
> The goal of EraseFlow is straightforward: make the target prompt trajectory behave exactly like a safe anchor prompt trajectory. Because we sample every trajectory from this anchor, all trajectories are “safe” by design, so we never have to score or inspect those from the unsafe prompt. The reward can therefore be held constant—it merely normalizes the trajectory-balance loss rather than steering it. Minimizing this loss on anchor-derived trajectories is enough to copy the anchor’s behavior onto the target prompt, achieving concept erasure without any separate reward model. This also aligns with the next query below. We will add this clearer explanation to the main paper.
>
> ### Constant vs. Learned reward.
>
> We have provided this ablation in Figure (3) of the main paper, where we specifically analyzed the effect of the NSFW classifier as the reward model. We observed that TB with learned reward itself is better than the DB objective. However, constant reward further improves the performance of TB objectives.
> Importantly, we observed that the NSFW classifier introduced some adversarial effects into the system. If we take a look at our TB formulation more carefully, Eq. (8), we can see that the reward is calculated for the anchor trajectory, which is on the “safe” prompt. If we know it is a safe image, then there is no need for the learned reward model because it is safe all the time. Notably, this is not possible for standard RL alignment methods as the reward is being calculated for the potentially generated unsafe image.
>
> ### Extension to multiconcept erasure
> We scaled EraseFlow to progressively larger sets of artist concepts and evaluated performance using CLIP-based unlearning and retention scores, along with the trade-off metric $H_a$ = Retain − Unlearn. Across all settings, EraseFlow consistently forgot the target concepts while preserving overall image quality. We also introduced a small retaining-artists set and applied the same loss on anchor-prompt trajectories, ensuring removals stay focused without degrading other capabilities. Even as the task grew harder with more concepts, EraseFlow remained stable and controllable, demonstrating practical targeted erasure with minimal disruption. Full results and analysis will appear in the camera-ready version.
>
> | **Artist**   | **Unlearn &darr;** | **Retain &uarr;** | $H_a$ &uarr;  | **CLIP &uarr;** | **FID &darr;** |
> |--------------|-----------:|-----------:|--------:|---------:|--------:|
> | Artist 1     |      11.60 |      25.61 |   14.01 |    26.08 |   17.69 |
> | Artist 5     |      18.90 |      24.40 |    5.50 |    25.78 |   17.94 |
> | Artist 100   |      22.90 |      22.50 |   –0.40 |    25.07 |   18.73 |
>
>
>
> We hope these clarifications address all your concerns and enhance the comprehensiveness of our work. We look forward to the discussion.
>
>
>
> ---
>
> [1] Anh Bui, Trang Vu, Long Vuong, Trung Le, Paul Montague, Tamas Abraham, Junae Kim, & Dinh Phung. (2025). _Fantastic Targets for Concept Erasure in Diffusion Models and Where To Find Them_. arXiv:2501.18950.

---

> > ### Author Response · Authors · 2025-08-06
> >
> > Dear Reviewer,
> >
> > Thank you again for your kind review. We hope we have successfully resolved all the questions. We'd be happy to clarify any remaining questions and are available to answer any questions you may have. Looking forward to hearing your thoughts.
> >
> > Best,
> > Authors

---

> > ### Comment · Reviewer_QEG1 · 2025-08-07
> >
> > Thanks to the authors for the rebuttal. This addresses all the concerns I had. I will discuss with other reviewers and will update my rating as appropriate.

---

### Author Response · Authors · 2025-08-01
**Clarification Regarding Sharing Anonymous Code Link During Rebuttal**

Dear AC,

We would like to share an anonymous GitHub link ([https://anonymous.4open.science](https://anonymous.4open.science)) with reviewer (qUE2) to facilitate their review of our code. We will ensure strict anonymity as per the [author guidelines](https://neurips.cc/Conferences/2025/CallForPapers).

However, we could not find explicit guidance regarding sharing code link during the rebuttal period. Therefore, we wanted to seek your clarification and approval before proceeding.

Thank you for your assistance.

Best regards,

Authors

---

### Note · Authors · 2025-08-15

We sincerely thank all the reviewers for their time and effort throughout the review process and their insightful reviews and feedback, which helped make our work more comprehensive. We are also glad that our clarifications and additional experimental results have **successfully addressed the questions and concerns in the original reviews.** It is gratifying to observe the **positive evaluations across various dimensions of our work,** as highlighted unanimously by the reviewers. We also thank the reviewer qUE2 for raising the scores and reviewer kTnT for considering an increase in the score.

We appreciate all the reviewers for their insightful suggestions on additional experiments and ablations. For example, reviewers QEG1, kTnT, and qUE2 particularly asked for various ablations to measure the robustness to hyperparameters sensitivity, where we showed that EraseFlow is consistent across the dimensions. Reviewer kTnT’s question also enabled us to highlight EraseFlow’s extensibility to state-of-the-art transformer-based architectures such as Flux. We further thank reviewer qUE2 for the thoughtful follow-up on the $\beta$ ablation, which motivated us to refine our intuitive explanation with stronger empirical evidence.

Finally, we are committed to further refining our work with new ablations, results, and discussions from the rebuttal phase for future readers. We also plan to open-source the entire codebase for reproducibility and to encourage the community to explore this direction.

---

### Decision · Program_Chairs · 2025-09-17

**Decision:**

Accept (spotlight)

**Comment:**

This paper introduces EraseFlow, a novel framework that formulates concept erasure in diffusion models as trajectory-level distribution matching using GFlowNets with trajectory balance. Reviewers agree that the work is original, technically sound, and well presented. The reviewers also appreciate strong empirical results showing improved trade-offs between erasure quality, prior preservation, and efficiency. While major concerns center on missing ablation studies (e.g., anchor prompt choice, hyperparameters), lack of experiments on multi-concept erasure, limited evaluation on other diffusion architectures, and incomplete reporting of computational/memory costs, they are well addressed during the rebuttal phase. The consensus is clearly above the acceptance threshold, and the AC agrees with the reviewers' decision.